# Can different variations of suspension exercises provide adequate loads and muscle activations for upper body training?

Faik Vural[1], Berkant Erman[2], Igor Ranisavljev[3], Yasin Yuzbasioglu[1], Nemanja Ćopić[4], Tolga Aksit[1]*, Milivoj Dopsaj[3,5], Mehmet Zeki Ozkol[1]

1 Coaching Education Department, Faculty of Sport Sciences, Ege University, Izmir, Turkiye, 2 Department of Sports and Health Sciences, Institution of Health Sciences, Ege University, Izmir, Turkiye, 3 University of Belgrade Faculty of Sportand PhysicalEducation, Belgrade, Serbia, 4 University "Union - Nikola Tesla", Faculty of Sport, Belgrade, Serbia, 5 Institute of Sport, Tourism and Service South Ural State University, Chelyabinsk, Russia

* tolga.aksit@ege.edu.tr

**Data Availability Statement:** All relevant data are within the paper.

## Abstract

The purpose of this study was to assess the differences in muscle activation (EMG) and body weight distribution (%BW) between suspension (TRX™ push-up and TRX™ inverted row) and conventional exercises (bench press and lying barbell row) using different contraction types (isometric and isotonic) and position variations (feet on the ground [FG] and feet on suspension device [FD]). It was also used to determine the intensity of the force applied to the straps of the suspension device corresponding to one repetition maximum (1-RM). Twelve male athletes (ages—24.5±4.2 years (mean±standard deviation [SD]); Height—181.0±6.8 cm; body mass—83.08±6.81 kg) participated in this study. Two suspension devices were used, one for the FD variation and one for the FG variation pectoralis major (PM) and triceps brachii (TRI) activations were assessed during the TRX™ push-up and bench press exercises. Transversus trapezius (TRA) and biceps brachii (BB) activations were assessed during the TRX™ inverted row and lying barbell row exercises. The results showed significant differences between exercises (FG and FD variations of TRX™ push-up and bench press) in PM activities (isometric and isotonic) ($p \leq 0.05$). However, these differences were only observed during isometric TRI activation ($p \leq 0.05$). In the FG and FD variations of the TRX™ inverted row and lying barbell row exercises, there were only differences in the isometric contractions of the TRA and BB ($p \leq 0.05$). In the suspension device of push-ups and inverted row for the FD variations, 70.5% and 72.64% of 1-RM intensity were obtained, respectively. Similar responses to training intensities and muscle activations can be obtained in suspension exercises and conventional exercises. FD variations of suspension exercises can be more effective in terms of muscle activations than FG variations, and isotonic suspension exercises increase exercise intensity more than isometric suspension exercises.

**Funding:** This study funded financially by Scientific Research Projects Coordination of Ege University (BAP-Project no:13-BESYO-002). The funders had no role in study design, data collection and analysis, decision to publish, or preparation of the manuscript.

**Competing interests:** The authors have declared that no competing interests exist.

## Introduction

In recent years, suspension exercises have become more popular than conventional exercises. Increased prime mover or core muscle activation is one of the most visible results of this exercise [1]. This may be considered a good strategy for both breaking up the monotony of conventional exercises and changing muscle activation on the basis of the training goals. Many additional exercises, such as push-ups, inverted rows, prone bridges, and hamstring curls, can be performed [1, 2].

The activation of prime movers and other muscle groups was reported to be much higher in suspension push-ups than in standard push-ups [1, 3, 4]. In addition, suspension exercises can be varied using devices such as that with a pulley [5]. While the pulley suspension device enhances instability, it also promotes muscular activation [1, 4]. However, this increase is not statistically significant in the some muscles [6]. As this example instability the core or synergist muscles may be activated more than the prime mover muscles in a training program, with severe instability

Suspension push-up exercises may also be performed with varying ground reaction forces and applied forces on the straps by altering the body inclination and strap length [6, 7]. When the angle of the TRX™ straps was adjusted to 90˚ toward the ceiling, 50.4% body weight (BW) was found in the elbow extension phase and 75.3% BW in the elbow flexion phase [8]. This was equivalent to the relative loads obtained in the flexion phase of the conventional push-up exercise. The relative load of conventional push-ups was determined to be 69% of BW during elbow extension, 75% during elbow flexion, and 49% in a knee-bent position [2, 9]. Furthermore, as the length of the straps of the suspension device increased (decreasing body inclination with respect to the ground), the loads on the straps increased, and the ground reaction forces decreased [7].

The inverted row is also a common suspension exercise. When the suspended inverted-row exercise was compared with the conventional inverted-row exercise, muscle activation was not significantly increased in prime movers[1, 10, 11]. In contrast, activation of the biceps brachii (BB) was shown to be greater during the suspension exercise than during the conventional exercise [1, 10, 11]. In this exercise, variations in muscle activation were more closely linked to changes in the body inclination and handgrip position. According to Snarr et al. [12], the activation of the posterior deltoid and middle trapezius is reduced with a supinated handgrip position because adopting a supine hand grip during elbow flexion improves biceps brachii recruitment [1].

Muscle activation is considered to occur as a result of unstable exercise regimens, which produce neuromuscular stress and enhance motor unit activation [13]. The height of the suspension device, gripping position, and angle between the body and ground affect this activation [1, 8]. Surface electromyographic (sEMG) responses have been reported to be expressed as loads [1, 14]. However, it is unclear how much load is produced on the basis of one repetition maximum (1-RM), suspension exercises with feet on the strap variation, and EMG changes during exercise. These are crucial factors to be considered when performing strength and conditioning activities. For this reason, in this study, measurements (i.e., EMG, applied forces on straps, and ground reaction forces) were obtained from 12 different types of exercises: a) feet on the ground (FG) and feet on the suspension device (FD) TRX™ push-up with isometric and isotonic contractions; b) FG and FD TRX™ inverted row with isometric and isotonic contractions; c) isometric and isotonic bench press (BP); and d) isometric and isotonic lying barbell row (LBR) exercises, TRX™ push-up, and TRX™ inverted row exercises compared with conventional exercises (BP and LBR).

The present study aimed to determine the percentage of 1-RM in conventional exercises to which applied forces in suspension exercises correspond and to compare the activation of

sEMG between suspension and conventional exercises. We hypothesized that i) the FD variation of suspension exercises may show a higher percentage of the 1-RM load than the FG variation and ii) the FD variation of suspension exercises may show a higher activation of sEMG than the FG variation.

## Materials and methods

### Participants

The present investigation was designed as a cross-sectional within-subject study. The sample size was calculated using G*Power 3.1.9 [power size (1-β) = 0.80, effect size f(V) = 0.97, type-1 error (α) = 0.05, group = 1, and measurements = 3]. According to power analsis the sample size was found to be 12 twelve. All participants have at least five years of track and field background male athletes (age—24.5±4.2 years (mean±standard deviation [SD]), height—181.0 ±6.8 cm, body mass—83.08±6.81 kg,) participated in this study. All participants had a history of resistance exercise for at least 1 year (1.35±0.2) and were right-handed. Each participant's height was measured using a stadiometer (Seca, Germany), with an accuracy of 0.1 cm. Anthropometric data were obtained using a body composition analysis device (Tanita BC 418, Tanita Corp., Tokyo, JPN), which uses the bioelectric impedance method. The inclusion criteria for this study were as follows: (a) no chronic limb discomfort or limitations that had a negative effect on exercises (e.g., back pain and lateral epicondylitis); (b) no serious injury or history of surgery in the past year (e.g., rotator cuff tear surgery); and (c) inclusion of the TRX™ inverted row, TRX™ push-up, LBR, and BP in the exercise program for at least 1 year. All participants voluntarily provided written informed consent before participating in the study. The experimental procedures were conducted in accordance with the principles of the Declaration of Helsinki [15].

### Instrumentation and data processing

sEMG for muscle activity was performed using a wireless EMG amplifier (Bionomadix, BN-EMG2, Biopac Systems, Inc., Goleta, USA) integrated with the Biopac MP150 data acquisition and analysis system (Biopac Systems, Inc., Goleta, USA), which is capable of 4-input channel EMG. The monitoring system recorded all EMG at a sampling rate of 1.000 Hz, and cut-off frequencies of 20–500 Hz band-pass Butterworth filter were applied and analyzed via Acq-Knowledge 4.2 software (Biopac Systems, Inc., Goleta, CA, USA). Disposable Ag–AgCl surface electrodes (Karditek Electrode, Karditek Medical Devices Co. Ltd., Izmir, TUR) were used during measurements. Before placing the surface electrodes, all skin areas were shaved and cleaned with sterile cotton and alcohol to reduce impedance. All electrodes were placed on the right side (dominant) of the participants and on the belly of the PM and triceps brachii (TRI), transverse trapezius (TRA), and BB following the SENIAM (Surface Electromyography for the Non-Invasive Assessment of Muscles project which was recommended for sensor placement procedures and signal processing methods for sEMG) method [16]. PM and TRI activities were recorded for variations (isometric and isotonic) of TRX™ push-up and isometric and isotonic BP exercises. TRA and BB activities were recorded for variations (isometric and isotonic) of the TRX™ inverted row and isometric and isotonic LBR exercises. The TRX™ push-up and inverted row exercises were performed in two ways: with FG and with FD. Two "S" type load-cells (SBS1 load-cell, CAS Electronic Industry, and Trade Inc., Korea) were placed on the straps to obtain applied forces from the arms. Each load cell was connected to the data acquisition system (BIOPAC MP150, Biopac Systems, Goleta, CA, USA). Data were recorded continuously at 1000 Hz using the AcqKnowledge 4.2 software (Biopac Systems, Inc., Goleta, CA, USA). A standard Olympic bar (19.6 kg) was used for BP and LBR exercises.

### Exercise sessions and applications

This study was carried out in four sessions in a laboratory at 72-h intervals. The exercise tempo was set at 60 bpm (1:0:1:0), which signified a one-count eccentric motion followed immediately by a one-count concentric motion, without any rest count before the next repetition. For the tempo setting, a digital metronome was use. The first session aimed to familiarize the participants with the measurement procedures and exercise protocols (TRX™ inverted row, TRX™ push up, LBR, and BP). Participants who could not adapt to the metronome and had extreme difficulty in performing the exercises were excluded from the study, and there was no need for any participant to exclude the study. In the second session, the maximal isometric voluntary contraction (MVIC) in the morning (09:00–12:00) and 1-RM in the afternoon (13:00–16:30) were determined. In the third session, the participants performed the suspension exercises (Fig 1) in the morning (09:00–11:00) and afternoon (14:00–16:00). In the fourth session, participants performed isometric and isotonic BP exercises (Fig 1D) and isometric and isotonic LBR (Fig 1A) exercises with the same loads as those of the previous session (described below). When performing the suspension exercises, the angle of the TRX™ straps was set at 90˚ toward the ceiling. During elbow extension of the TRX™ inverted row and elbow flexion of the isotonic TRX™ push-up exercise, the body position of the participants was tried to keep in a neutral position with head, trunk, and hip straight line) with FG (Fig 1C–1E) and FD (Fig 1B–1F) during exercise. Strap lengths of the devices were adjusted according to this evaluation. The elbow flexion angle was 90˚ during the isometric TRX™ push-up, TRX™ inverted row, and BP exercises. However, no limitations were applied to the elbow flexion angle in other isotonic exercises. Therefore, the closest comparison between the conventional (BP and LBR) and suspension (TRX™ push-ups and TRX™ inverted row) exercises could be made. Participants performed inverted row exercises shoulder width apart and with pronated grips.

In the first laboratory session (familiarization), the participants completed 10-min individual warmups. The metronome was started (60 bpm, 1:0:1:0), and they performed the TRX™ inverted row (FG and FD), TRX™ push-up (FG and FD), LBR, and BP exercises with electrodes

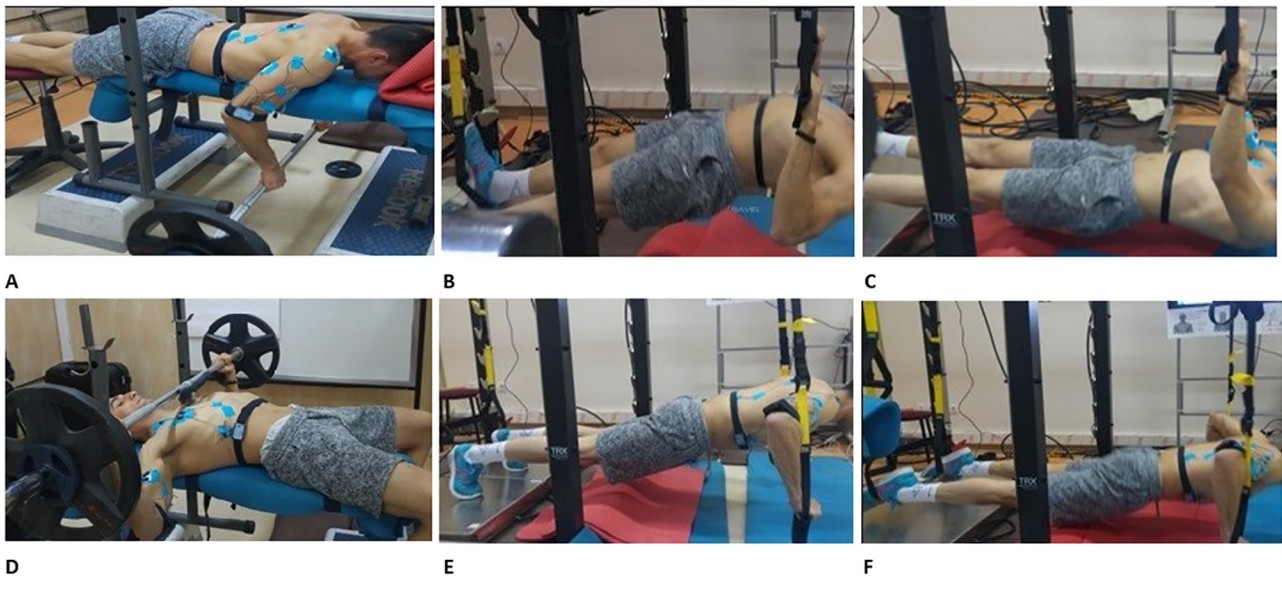

**Fig 1. Application of exercises.**

until the participant could adapt to the metronome. Individual repetitions were performed for each exercise, and a 5-min rest was applied between the exercises. After 72 h, participants visited the laboratory twice during the second session—in the morning (09:00–12:00) and afternoon (14:00–16:30)—to determine MVIC and 1-RM.

*MVIC and 1-RM Determination*; In the morning, MVICs were determined according to the protocols established by Konrad [17]. The PM and TRI muscles were measured during BP exercises. A manual goniometer was used in the study. Elbow angle was tried to fix at 90˚ during measurements. After a low-intensity 10-min warm up, participants exercised with maximum effort for 3 s on verbal instructions from the researchers, which was repeated three times, with 60-s pauses between efforts [17]. Therefore, this application allowed for the normalization of the EMG. These procedures were performed for each muscle, and the repetition with the maximal effort among the three repetitions was considered for evaluation. The electrodes for PM measurements were placed in the following manner: four fingers below the clavicle, at the midpoint between the sternal notch and the anterior axillary line. For the MVIC measurement, the participants were placed in the supine position on a flat bench with the elbows at 90˚. To prevent movement of the trunk, it was fixed on the bench, and the participants were asked to push the fixed bar adapted to the bench with the palm of the hand with maximal thrust. Muscle activations were recorded during this sequence. The electrodes for TRI measurements were placed between the acromion and olecranon at the midpoint of the posterior region of the upper arm. For the MVIC measurement, the participants placed their hand on a fixed hanging mechanism with the elbows at 90˚ and the palm facing the ground. Subsequently, three repetitions were performed by the participants who pressed the fixed handle down with the palm to create maximum resistance, and the muscle activations during this period were recorded. The electrodes for TRA measurements were placed at the midpoint of the horizontal line between the vertebral root of the scapula and third thoracic spinous process. For MVIC measurements, the participants were placed in a sitting position to prevent the body from moving forward. During this time, the participants were asked to hold the handle adapted to a fixed point opposite the seat, with the arm in a 90˚ abduction and internal rotation position. Subsequently, the participants were asked to pull the fixed handle horizontally toward themselves to create maximal resistance with their palm. Muscle activations were recorded at this time. The electrodes for BB measurements were placed at the midpoint above the short head of the biceps brachii at 2 cm from the proximal tendon to the distal tendon. For MVIC measurements, the participants were placed in a sitting position with the elbow bent at 90˚, the forearm in supination (palm towards the face) and standing upright. Subsequently, the participants pulled the fixed handle towards themselves with their palm to create maximum resistance. Three repetitive measurements were taken, and the muscle activations during this period were recorded. In the afternoon, the 1-RM determination was calculated using the indirect method and the formula 1-RM (kg) = Weight/[1.0278-(0.0278×Repetitions)] [18]. Therefore, normalization could be performed for the applied forces of the 1-RM in the suspension exercises.

After 72 h, participants visited the laboratory for the third session in the morning (09:00–11:00) and afternoon (14:00–16:00). TRX™ push up (morning session) and inverted row (afternoon session) exercises were performed in two variations: the FG and FD. Firstly, a 10-s isometric FG variation of TRX™ push up [TRX™ push up$_{(isom\_FG)}$] was performed for two sets of 10 s, with a 2-min rest between the sets (10 s × two sets, with 2-min rest). In 10-s isometric exercises, EMG and load-cell data showed a long plateau, and the 3-s part of this plateau was averaged for evaluation. The first and the last 3 seconds are excluded from the data to prevent the data from being excluded by the small oscillations that may occur in the body. After a 10-min rest, a five-rep isotonic FG variation of TRX™ push up [TRX™ push up$_{(isot\_FG)}$] was

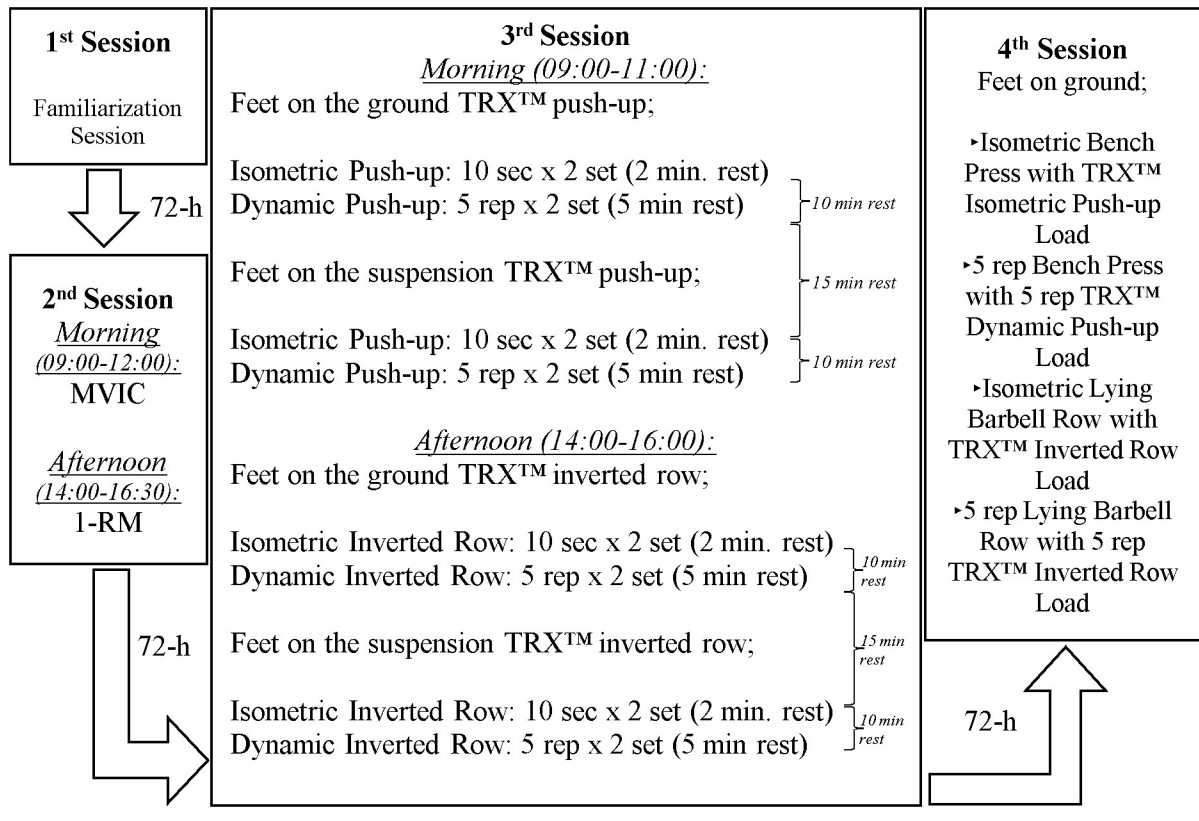

**Fig 2. Application of sessions.**

performed (five reps × two sets, with 5-min rest). The average EMG and load-cell data were calculated by extracting the first and last repetitions from five repetitions, and the data of the remaining three repetitions were averaged for the evaluation of the isotonic exercises. After a 15-min rest, the FD variation of TRX™ push up [TRX™ push up$_{(isot\_FD)}$] was performed using the same protocols (Fig 2). The participants visited the laboratory in the afternoon for the FG and FD variations of the TRX™ inverted row exercises. The same method was used for the morning exercise procedure (Fig 2). In the fourth session (after 72 h), isometric–isotonic BP [BP$_{(isot, isom)}$] and LBR [LBR$_{(isot, isom)}$] exercises were performed. In these exercises, loads obtained from the third session of TRX™ exercises were used. For example, if the participant applied a total of 40 kg (right and left arms) of strap force for the TRX™ push-up$_{(isom\_FG)}$ exercise in the third session, in the next (fourth) session, the participant performed isometric BP exercise [BP$_{(isom)}$] with 40 kg (same as that obtained from TRX™ isometric push-up load, Fig 2); the same procedures were applied for the isotonic exercises. Therefore, EMG variables could be compared between conventional and suspension exercises, and the percentages of the applied forces corresponding to %1-RM and %BW were determined.

## Statistical analysis

Muscle activation and force measurements with the suspension devices were obtained from four different types of exercise variations: isometric (10 s), isotonic (five repetition) and FG (with two straps), and FD (with four straps). The applied forces on the straps were normalized to the BW distribution (%BW) and 1-RM (%1-RM). EMG variables were normalized using

MVIC (%MVIC). Data are presented as means and standard deviations. To assess the differences in EMG between different exercise types for the same muscle in three different exercises [e.g., isometric (10 s) pectoralis major activation in the FG variation of TRX™ push-up, FD variation of TRX™ push-up, and BP], one-way repeated measures analysis of variance (ANOVA) was performed. Partial eta squared ($\eta_\rho^2$) was used as the effect size and categorized as small (0.01–0.06), medium (0.06–0.14), or large (> 0.14) [19]. The Holm–Bonferroni adjustment was used for multiple comparisons. A paired sample t-test was performed to assess the differences between exercises (isometric vs. isotonic and FG vs. FD). Cohen's d was used as the effect size and categorized as no effect (0–0.2), small effect (0.2–0.5), medium effect (0.5–0.8), and large effect (>0.8) [20]. Pearson's correlation coefficients (r) were used to determine the relationship between %MVIC and %1-RM and %MVIC and %BW. Data analysis was performed using SPSS software (IBM SPSS Statistics for Windows, Version 25.0. Armonk, NY: IBM Corp). The statistical significance level was set at P ≤ 0.05.

## Results

The Participants characteristics with 1-RM and MVIC (M±SD) variables was shown in Table 1.

TRA and BB activations were used for TRX™ inverted and LBR exercises. The one-way repeated ANOVA results showed significant differences between the exercises (FG and FD variations of TRX™ push-up and BP) in PM activities (isometric and isotonic contractions). The FD variation of TRX™ push-up was greater than the FG and conventional BP. However, these significant differences were only found in the isometric TRI activation (p≤0.05) (Table 2). In TRX™ inverted row and LBR exercises, there were only significant differences between the exercises in isometric TRA and isotonic BB (p≤0.05) (Table 3). The FG variation of TRX™ inverted rows was greater than FD and conventional LBR exercises. The %MVIC and comparisons for muscle activities during exercises are also shown in Tables 2 and 3.

In terms of %1-RM and %BW, there were significant differences between isometric and isotonic of TRX™ push-ups and TRX™ inverted rows during FG and FD (p<0.001, 0.72≤ES≤1.87) (Table 4). In terms of 1-RM, the isometric FG variation was greater than the isotonic exercise, and the isotonic FD variation of the push-up was greater than the isometric push-up. There were also significant differences between the FG and FD of TRX™ push-up and TRX™ inverted row exercises (p<0.001,1.15<ES<4.68) (Table 5).

Bivariate correlations between applied forces (%1-RM, %BW) and muscle activity (%MVIC) are presented in Table 6. A significant negative relationship was found between TRI

**Table 1. Participants characteristics with 1-RM and MVIC (M±SD).**

| | |
|---|---|
| Ages (years) | 24.5±4.2 |
| Height (cm) | 181.0 ± 6.8 |
| Body mass (kg) | 83.08 ± 6.81 |
| 1-RM—Bench Press (kg) | 101.75 ± 15.88 |
| 1-RM—Lying Barbell Row (kg) | 88.75 ± 13.28 |
| MVIC—PM (mV) | 0.12 ± 0.06 |
| MVIC—TRI (mV) | 0.22 ± 0.11 |
| MVIC—TRA (mV) | 0.17 ± 0.06 |
| MVIC—BB (mV) | 0.48 ± 0.1 |

1-RM: One Repetition Maximum; MVIC: Maximum Voluntary Isometric Contraction; PM: Pectoralis Major; TRI: Triceps Brachii; TRA: Transverse Trapezius; BB: Biceps Brachii.

**Table 2. The one-way repeated ANOVA results and multiple comparisons for suspension push-ups and bench press exercises.**

| Measurement | | $F_{(2,22)}$ | Repeated ANOVA $p$ | $\eta_p^2$ | Type of Exercise (M±SD) | Holm-Bonferroni $p$ |
|---|---|---|---|---|---|---|
| | | | | | **FG- FD** | |
| | | | | | **FG- BP** | |
| | | | | | **FD- BP** | |
| Isometric | PM %MVC | 4.353 | 0.058 | 0.284 | 50.56±24.5–103.3±82.8 | 0.031* |
| | | | | | 50.56±24.5–49.52±28.69 | 1.0 |
| | | | | | 103.3±82.8–49.52±28.69 | 0.27 |
| | TRI %MVC | 10.235 | 0.001[†] | 0.482 | 16.19±2.9–25.08±8.8 | 0.024* |
| | | | | | 16.19±2.9–15.83±4.9 | 1.0 |
| | | | | | 25.08±8.8–15.83±4.9 | 0.021* |
| Isotonic | PM %MVC | 18.727 | <0.001[†] | 0.630 | 96.3±48.2–121.2±53.2 | <0.001[†] |
| | | | | | 96.3±48.2–85.7±32.7 | 0.32 |
| | | | | | 121±53–85.7±32.7 | 0.002* |
| | TRI %MVC | 2.288 | 0.125 | 0.172 | 37.4±7.3–47.08±18.5 | 0.12 |
| | | | | | 37.4±7.3–39.97±26.76 | 1.01 |
| | | | | | 47.08±18.5–39±26 | 0.081 |

*$p \leq 0.05$;

[†] $p \leq 0.001$;

$\eta_p^2$: partial eta effect size; M±SD: Mean ± Standard Deviation; Mean Diff.: Mean difference; %MVC: Percentage of maximum voluntarily isometric contraction; PM: pectoralis major, TRI: Triceps brachii; FG: Feet on the ground during TRX™ Push-up; FD: Feet on the suspension device during TRX™ Push-up; BP: Bench press

**Table 3. The one-way repeated ANOVA results and multiple comparisons for suspension inverted rows and lying barbell row exercises.**

| Measurement | | $F_{(2,22)}$ | Repeated ANOVA $p$ | $\eta_p^2$ | Type of Exercise (M±SD) | Holm-Bonferroni $p$ |
|---|---|---|---|---|---|---|
| | | | | | **FG- FD** | |
| | | | | | **FG- LBR** | |
| | | | | | **FD- LBR** | |
| Isometric | TRA %MVC | 6.583 | 0.005* | 0.384 | 67.14±8.57–53.22±12.1 | 0.009* |
| | | | | | 67.14±8.57–48.14±13.8 | 0.03* |
| | | | | | 53.22±12.1–48.14±13.8 | 1.0 |
| | BB %MVC | 0.410 | 0.668 | 0.036 | 24.54±9.7–27.75±15.34 | 0.52 |
| | | | | | 24.54±9.7–28.29±16.56 | 1.0 |
| | | | | | 27.75±15.34–28.29±16.56 | 1.0 |
| Isotonic | TRA %MVC | 0.818 | 0.450 | 0.069 | 67.64±19.97–61.13±8.5 | 0.44 |
| | | | | | 67.64±19.97–59.88±20.7 | 1.0 |
| | | | | | 61.13±8.5–59.88±20.7 | 1.0 |
| | BB %MVC | 11.526 | <0.001[†] | 0.512 | 33.27±8.7–28.86±10.06 | <0.001[†] |
| | | | | | 33.27±8.7–30.51±11.41 | 0.136 |
| | | | | | 28.86±10.06–30.51±11.41 | 0.131 |

*$p \leq 0.05$;

[†] $p \leq 0.001$;

$\eta_p^2$: partial eta effect size; M±SD: Mean ± Standard deviation; Mean Diff.: Mean difference; %MVC: Percentage of maximum voluntarily isometric contraction; TRA: Transverse trapezius; BB: Biceps brachii; FG: Feet on the ground during TRX™ Inverted Row; FD: Feet on the suspension device during TRX™ Inverted Row; LBR: Lying barbell row.

**Table 4. Paired sample t-test results for the percentage of one repetition maximum and body weight of isometric and isotonic exercises.**

| Exercise | | Isometric vs Isotonic (% 1-RM) M±SD | p | ES | Isometric vs Isotonic (Arms % BW) M±SD | p | ES | Isometric vs Isotonic (Legs % BW) M±SD | p | ES |
|---|---|---|---|---|---|---|---|---|---|---|
| TRX™ Push-up | FG | 61±6.1 vs 59.0±6.5 | <0.001† | 1.44 | 72.39±1.62 vs 70.11±1.52 | 0.001† | 1.33 | 27.60±1.62 vs 29.88±1.52 | 0.001† | 1.33 |
| | FD | 63.5±6.9 vs 70.5±5.9 | <0.001† | 1.67 | 74.96±1.62 vs 72.08±1.19 | <0.001† | 1.37 | 25.03±1.62 vs 27.19±1.19 | <0.001† | 1.37 |
| TRX™ Inverted Row | FG | 66.5±5.3 vs 70.5±5.9 | <0.001† | 1.87 | 69.49±2.97 vs 73.38±1.85 | <0.001† | 1.34 | 30.50±2.97 vs 26.61±1.85 | <0.001† | 1.34 |
| | FD | 70.19±4.6 vs 72.64±3.8 | 0.029* | 0.72 | 73.33±2.85 vs 75.69±3.04 | 0.001† | 0.88 | 26.66±2.85 vs 24.30±3.04 | 0.001† | 0.88 |

*p≤ 0.05;

† p≤ 0.001;

Mean±SD: Mean ± Standard deviation; ES: Effect size; FG: Feet on the ground during exercise; FD: Feet on the suspension device during exercise; %1-RM: Percentage of one repetition maximum; %BW: Percentage of body weight distribution.

activity %MVIC and %1-RM in the TRX™ push-up$_{(isom\_FG)}$ (p<0.005) (Table 5). There were also significant negative relationships between PM activity and applied forces in the TRX™ push-up$_{(isom\_FG)}$ (p<0.005) (Table 6). There was no other significant relationship between applied forces and muscle activations (Table 6).

## Discussion

TRX™ push-ups and TRX™ inverted rows were used for suspension exercises, and BP and LBR were used as conventional exercises in the present study. The PM, TRI, BB, and TRA muscles were studied in several suspension and conventional exercises. In addition, the study investigated the amount of force applied to the straps by the hands equivalent to a percentage of 1-RM and how BW was distributed in suspension exercise variations during the suspension exercise session.

The PM is one of the prime mover muscle groups in the TRX™ push-up exercise. Consequently, EMG responses of this muscle have been widely examined in the literature [4, 5, 21].

**Table 5. Paired sample t-test results for feet on the ground and feet on the suspension device of TRX™ push-up and inverted exercises.**

| Exercise | | FG vs FD Mean Diff. | $t_{(11)}$ | p | ES |
|---|---|---|---|---|---|
| TRX™ Push-up | Isometric (%1-RM) | -0.0252 | -6.295 | <0.001† | 1.44 |
| | Isotonic (%1-RM) | -0.025 | -7.542 | <0.001† | 1.23 |
| | Isometric (Legs %BW) | -2.566 | -16.261 | <0.001† | 4.68 |
| | Isotonic (Legs %BW) | -2.690 | -10.592 | <0.001† | 4.67 |
| | Isometric (Arms %BW) | 2.566 | 16.261 | <0.001† | 4.68 |
| | Isotonic (Arms %BW) | 2.690 | 10.592 | <0.001† | 4.67 |
| TRX™ Inverted Row | Isometric (%1-RM) | -0.040 | -6.925 | <0.001† | 1.97 |
| | Isotonic (%1-RM) | -0.024 | -5.830 | <0.001† | 1.37 |
| | Isometric (Legs %BW) | -3.836 | -6.544 | <0.001† | 2.18 |
| | Isotonic (Legs %BW) | -2.314 | -2.703 | 0.021* | 0.78 |
| | Isometric (Arms %BW) | 3.836 | 6.544 | <0.001† | 2.18 |
| | Isotonic (Arms %BW) | 2.314 | 2.703 | 0.021* | 0.78 |

*p≤ 0.05;

† p≤ 0.001;

Mean Diff: Mean differences; ES: Effect size; FG: Feet on the ground during exercise; FD: Feet on the suspension device during exercise; %1-RM: Percentage of one repetition maximum; %BW: Percentage of body weight distribution.

**Table 6. Bivariate correlation between percentage of one repetition maximum and body weight and muscle activations.**

| Measurement | | | Pectoralis Major (%MVC) | Triceps Brachii (% MVC) | Transverse Trapezius (% MVC) | Biceps Brachii (% MVC) |
|---|---|---|---|---|---|---|
| Isometric TRX™ Push-Up | FG (%1-RM) | r | 0.582* | -0.597* | | |
| | | p | 0.047 | 0.040 | | |
| | FD (%1-RM) | r | 0.490 | 0.300 | | |
| | | p | 0.106 | 0.344 | | |
| | FG (Arms % BW) | r | -0.914* | 0.632* | | |
| | | p | <0.001 | 0.028 | | |
| | FD (Arms % BW) | r | -0.931* | 0.647* | | |
| | | p | <0.001 | 0.023 | | |
| Isotonic TRX™ Push-Up | FG (%1-RM) | r | 0.503 | 0.088 | | |
| | | p | 0.096 | 0.786 | | |
| | FD (%1-RM) | r | 0.419 | 0.555 | | |
| | | p | 0.175 | 0.061 | | |
| | FG (Arms % BW) | r | 0.467 | -0.671 | | |
| | | p | 0.126 | 0.017 | | |
| | FD (Arms % BW) | r | -0.640* | 0.550 | | |
| | | p | 0.025 | 0.064 | | |
| Isometric TRX™ Inverted Row | FG (%1-RM) | r | | | 0.722 | 0.183 |
| | | p | | | 0.008* | 0.569 |
| | FD (%1-RM) | r | | | -0.421 | 0.017 |
| | | p | | | 0.173 | 0.958 |
| | FG (Arms % BW) | r | | | 0.317 | 0.309 |
| | | p | | | 0.315 | 0.329 |
| | FD (Arms % BW) | r | | | 0.186 | -0.879* |
| | | p | | | 0.563 | <0.001 |
| Isotonic TRX™ Inverted Row | FG (%1-RM) | r | | | 0.488 | 0.261 |
| | | p | | | 0.108 | 0.447 |
| | FD (%1-RM) | r | | | 0.370 | 0.353 |
| | | p | | | 0.237 | 0.260 |
| | FG (Arms % BW) | r | | | 0.415 | -0.360 |
| | | p | | | 0.180 | 0.250 |
| | FD (Arms % BW) | r | | | -0.715* | -0.689* |
| | | p | | | 0.009 | 0.013 |

*p≤ 0.05;

FG: Feet on the ground during suspension exercise; FD: Feet on the suspension device during suspension exercise; %1-RM: Percentage of one repetition maximum; % MVC: Percentage of maximum voluntarily isometric contraction; %BW: Percentage of body weight distribution.

In an isometric BP exercise, Clemons and Aaron [22] found that PM activation had a 75% MVIC. However, PM activation in isometric BP was observed as ~50% in the present study. In the FD, the isometric variation of the TRX™ push-up showed ~100% more MVIC activation than the BP and the FG variation of the isometric TRX™ push-up. In this case, if participants want to increase the activation of the PM muscle for isometric exercise, the FD variation of the TRX™ push-up may be preferred to the conventional BP and the FG variation of the TRX™ push-up. In the FD variation of isotonic push-ups, the body moved forward toward the straps and the stabilization decreased further, and PM activation was observed to be higher than that of the BP, which is a more stable exercise. Simultaneously, this situation led to the difference

between FD and BP being greater than that between FD and FG. However, no difference was observed between the TRX™ pulley exercise (which has the least amount of stability) and the standard push-up [5]. The increase in instability and difficulty of movement in the pulley [5] and the TRX™ push-up$_{(isot\_FG)}$ in the present study does not imply that EMG responses of prime mover muscles will show significantly different activities with conventional exercise. The data revealed that the FD appears to be the critical component that produces the difference with conventional exercise.

Notably, core strength, which was not considered in this study, might be a key factor in the FD TRX™ push-up variation. The absence of core muscle EMG data may be considered one of the limitations of this study. Calatayud and Borreani [5] found that when using a suspension device with a pulley system, the rectus abdominis and erector lumbar spinae were activated more than they were when using the standard TRX™.

Higher activation of the TRI in isotonic conditions was observed with the isometric application in the FD version of the TRX™ push-up$_{(isom\_FD)}$, indicating that TRI activation played a more active role in providing stability. Suspension exercises are also known to affect the core muscles [23, 24], which are sometimes isometric, and have positively affected individuals with lower back pain complaints [23]. Although this was not monitored in this study, participants' trunks did not show any up-and-down movement and were, therefore, acceptable for the push-up. Thus, the greater enhancement of TRI activation appears to be compensated for by less stabilization of movement. The FD suspension exercise variation may be preferred to increase muscle activation of the PM and TRI. In addition, TRI activation increased more than PM activation when the contraction type changed from isometric to isotonic.

The percentage of 1-RM used by individuals throughout the suspension exercise was also determined. Therefore, it was easier to make comparisons between conventional BP and LBR exercises. In the FD with isotonic variation, 70% and 72% loads were obtained from TRX™ push-up and TRX™ inverted rows, respectively. This intensity level is commonly used to promote hypertrophy [25]. Considering that participants' relative strength for BP was ~122% (a lifting ratio of 1.22 times the participants' mean BW) and for LBR was ~106%, the FD TRX™ push-up and the FD TRX™ inverted row may be advised to promote muscular activation, at least for those with this strength.

When BW distribution (%BW) was evaluated during the exercises, the increase in instability was in the direction of the decrease in the load of %BW on the arms. In this case, the increase in synergistic muscle activation may have provided further evidence, along with the demonstration of %BW. In addition, 72% of %BW and 74% of %BW in the straps were obtained during the FG and FD variations of TRX™ push-ups, respectively. However, %BW was higher during isotonic exercise than during isometric exercise on straps. Isotonic exercise may have increased the load on the legs by increasing body oscillation. Giancotti et al. [7] also found that an exercise set up with a 238-cm suspension device strap length and body inclination of 20.8˚ to the ground resulted in 58% BW on the straps when the elbow was flexed. During the flexion phase in the present study, the participants' bodies were parallel to the ground, which was most likely the reason for the higher %BW. In addition, the %BW increase in the arms and the PM activation did not move in the same direction; nevertheless, the TRI muscle group moved in the same direction as %BW. Although %BW increases as elevation from the ground decreases [7], Borreani et al. [6] found that %MVIC obtained from heights of 10 cm and 65 cm to the ground for TRI showed the opposite. TRI activity was 27.6% higher at 65 cm than at 10 cm. More consideration should be given to the notion that a higher %BW distribution in the arms does not always accurately describe muscle activation.

In the isometric LBR exercise, the highest activation was obtained in the FG variation in FG and FD suspension exercises, and the lowest activation was observed in the conventional LBR

exercise. In isometric and isotonic exercises, TRA activation was almost unchanged in the FD variation compared with the FG one, with only a 0.74% difference. This may have caused the activation of the synergist muscle groups to increase further. However, since the core muscles were not observed in this study, it is difficult to make inferences. The TRA activation in the FD increased more than the BB activation when the exercise method changed from isometric to isotonic. Therefore, FD and reduced stabilization did not result in a difference in the TRA activation. However, BB activation was found to be higher in the suspension inverted row than in the conventional inverted row [1, 10, 11]. In addition, the FG variation may be preferable to the FD variation for BB activation. Moreover, the increase in %BW on the straps in the FD variation may have also increased the BB muscle activation.

In push-ups, the activation of the PM and TRI was greatest in the FD variation, which was less stable than FG. However, while reduced stability enhanced BB activation during the isometric exercise, it lowered BB activation during the isotonic exercise in inverted-row exercises. Moreover, the TRA did not exhibit the same tendency. In both contraction types, TRA activation was lower in the FD variants than in the FG variants. Compared with isometric contraction, during isotonic contraction, TRA activation increased more than BB activation. Isotonic exercise showed greater activation of both muscles in the FG variation.

According to this study, *(i)* with suspension exercises, similar responses to training intensities and muscle activations as those of conventional exercises can be obtained, *(ii)* the FD variation of suspension exercises can be more effective than the FG variation, and *(iii)* the isotonic type of suspension exercise increases exercise intensity more than the isometric type.

## Practical applications

If the individuals' relative strength ratio is 1.22 or more for BP and 1.06 or more for LBR exercises, the FD variation (individuals' body parallel to the ground in the flexion phase for push-ups and in the extension phase for inverted rows) may achieve more activation in the PM, TRI, BB, and TRA muscles than the FG variation. Although the TRI, BB, and TRA muscles showed greater activation for the FD variation during isotonic contraction, no significant difference was observed between this and the conventional variation. For this reason, performing a more challenging movement (e.g., more unstable) does not mean that more activation will be achieved than that of a conventional variation.

## Acknowledgments

We would like to express our gratitude to Dr. Ekim Pekunlu, who was with us at the start of the research but unfortunately passed away. Authors also gratefully thank to participants who involved to this study.

## Author Contributions

**Conceptualization:** Mehmet Zeki Ozkol.

**Data curation:** Faik Vural, Berkant Erman, Yasin Yuzbasioglu, Tolga Aksit.

**Investigation:** Faik Vural, Yasin Yuzbasioglu, Tolga Aksit.

**Methodology:** Mehmet Zeki Ozkol.

**Project administration:** Mehmet Zeki Ozkol.

**Resources:** Tolga Aksit, Mehmet Zeki Ozkol.

**Validation:** Berkant Erman, Milivoj Dopsaj.

Writing – **original draft:** Berkant Erman, Mehmet Zeki Ozkol.

Writing – **review & editing:** Igor Ranisavljev, Nemanja Ćopić, Tolga Aksit, Milivoj Dopsaj.

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
