## [Decision Letter · Decision Letter 0]

7 Sep 2022

PONE-D-22-17523Can Different Variation of Suspension Exercises Provide Adequate Loads and Muscle Activations for Upper Body Training?PLOS ONE

Dear Dr. Akşit,

Thank you for submitting your manuscript to PLOS ONE. After careful consideration, we feel that it has merit but does not fully meet PLOS ONE’s publication criteria as it currently stands. Therefore, we invite you to submit a revised version of the manuscript that addresses the points raised during the review process. Please submit your revised manuscript by Oct 22 2022 11:59PM. If you will need more time than this to complete your revisions, please reply to this message or contact the journal office at plosone@plos.org. Please include the following items when submitting your revised manuscript:A rebuttal letter that responds to each point raised by the academic editor and reviewer(s). You should upload this letter as a separate file labeled 'Response to Reviewers'.A marked-up copy of your manuscript that highlights changes made to the original version. You should upload this as a separate file labeled 'Revised Manuscript with Track Changes'.An unmarked version of your revised paper without tracked changes. You should upload this as a separate file labeled 'Manuscript'.

We look forward to receiving your revised manuscript.

Kind regards,

Xin Ye, Ph.D.

Academic Editor

PLOS ONE

Journal Requirements:

 "This study funded by Ege University Office of Scientific Research Projects (BAP-Project no:13-BESYO-002)."

Reviewers' comments:

Reviewer's Responses to Questions

**Comments to the Author**

1. Is the manuscript technically sound, and do the data support the conclusions?

Reviewer #1: Yes

Reviewer #2: Partly

2. Has the statistical analysis been performed appropriately and rigorously? 

Reviewer #1: Yes

Reviewer #2: Yes

3. Have the authors made all data underlying the findings in their manuscript fully available?

Reviewer #1: Yes

Reviewer #2: Yes

4. Is the manuscript presented in an intelligible fashion and written in standard English?

Reviewer #1: No

Reviewer #2: No

5. Review Comments to the Author

Reviewer #1: This article is very interesting and was rigorously carried out. However, the poor English grammar made the evaluation process very hard. I will not detail the areas of poor English because the list will be extensive. Moreover, it seems multiple authors wrote different part because the style of writing is different in the different areas of the manuscript.

Overall, this article will benefit from pictures of figures of the exercises use to give the reader the idea behind all the methodology and explanation of exercises.

Abstract, Introduction, Methods:

These section are full of grammatical issues making it hard to read and follow. I get the idea, background, and methodology used based on experience but probably most readers won't be able to follow or understand.

Results:

Overall, in addition to the poor English grammar having all values, numbers, and calculation make it even worst to follow. I would recommend to briefly write overall results and cite the tables included in the manuscript. The table are very well done and very explicit. Therefore, easy to follow.

Discussion:

Once again, in addition to the poor grammar most of the discussion is a repetition of results without minimal discussion of findings. This is very evident in the first 3 paragraphs. After the 3rd paragraph explanations are way better.

Reviewer #2: ABSTRUCT

Line 29, should be one decimal for height (181.x), not just 181. Also, please use “height”, not body height.

Line 38, differences (P≤0.05) between exercises, great! Which one is greater??

Line 41-42, the take-home message is incomplete and unclear. Which exercise is better? What individuals can be beneficial from this???

INTRODUCTION

General comments:

1) No hypotheses in the study.

2) The purpose of the study was not clear. Which 12 different types of exercises should be more specific.

Minor issues:

Line 50-52, many additional…..be performed. This is a run-on sentence.

Line 53, prime movers. Since the authors mentioned “much higher”, can the authors report % differences between suspension and standard push-ups in the references cited.

Line 60, [5] or (5)? Also, please check the format of this in-text citation.

Line 63-68, the current study did not examine oxygen consumption or RER, this paragraph is not necessary, please delete it.

Line 70-71, Differences ….variances. Run-on sentence again, please delete or complete the sentence.

Line 82, no significant increase in muscle activation (citations?).

METHODS

General comments:

1) I recommend the authors provide pictures of each exercise, that will help readers understand what exercises performed in the present study.

2) The authors should create a section for MVIC and 1-RM, not included in the “participants” section.

Minor issues:

Line 112, What did the authors mean recreationally athletes?? Athletes should be professional.

Line 113, height and one decimal.

Line 114, how did the authors measure height? Form BIA?

Line 120, the authors mentioned the participants perform TRX inverted row and push-up in their training programs. How many years of experience in performing TRX?

Line 128-129, MVIC and 1-RM on what exercises??

Line 130-131, what exercises did the participants perform in the third visit?

Line 147-148, on the belly of what muscles???

RUSULTS

General comments:

1) The authors should create a table for participants’ characteristics with MVIC, 1-RM, and training experience (years).

2) Whenever the authors mentioned differences, the authors should provide if there was a significant difference as well as relationship. Also, the authors should report which one was greater if there is a significant difference.

Minor issues:

Line 234, significant differences, great! Which one was greater?

Line 236, were the differences significant?

Line 238, again, were the differences significant? If yes, which one was greater?

Line 254 and 256, significantly differences, great! Which one was greater?

Line 271, found relationships…? Significant? Positive or negative??

DISCUSSION

General comments: The discussion is too long and the authors should create a paragraph to include take-home messages in the last section of the discussion.

PRACTICAL APPLICATIONS

LINE 430-431, what does this sentence mean? “it is possible to exercise with an intensity corresponding to 70% of 1-RM during isotonic suspension exercise.”

What are the authors trying to tell readers??

Line 433-434, I do not understand “further contribute to the future”??

Line 434, should be relative strength “ratio”?

The authors mentioned more unstable does not mean more activation than traditional variations? Why the authors mentioned may achieve more activations when the feet on the device variation?

6. PLOS authors have the option to publish the peer review history of their article (what does this mean?). If published, this will include your full peer review and any attached files.

Reviewer #1: No

Reviewer #2: No

---

## [Author Response · Author response to Decision Letter 0]

22 Oct 2022

First, we want to thank all reviewers for their time and grammar issues fixed in the “Manuscript” file. All changes can be seen in the “Revised” version. 

Response to the Reviewer 1#

“…. Overall, this article will benefit from pictures of figures of the exercises use to give the reader the idea behind all the methodology and explanation of exercises ….”

Response: Fig. 1 was added into the manuscript.

“…Abstract, Introduction, Methods: These sections are full of grammatical issues making it hard to read and follow. I get the idea, background, and methodology used based on experience but probably most readers won't be able to follow or understand…”

Response: Authors fixed grammar issues.

“…Results: Overall, in addition to the poor English grammar having all values, numbers, and calculation make it even worst to follow. I would recommend to briefly write overall results and cite the tables included in the manuscript. The table are very well done and very explicit. Therefore, easy to follow…”

Response: After all, “p<0.005”, we added cite of tables. Marked changes with “yellow” in the “Revised manuscript with track changes”

“….Discussion: Once again, in addition to the poor grammar most of the discussion is a repetition of results without minimal discussion of findings. This is very evident in the first 3 paragraphs. After the 3rd paragraph explanations are way better...”

Response: We fixed this part and marked in “Revised manuscript with track changes”

Response to the Reviewer 2#

Abstract:

1) Line 29, should be one decimal for height (181.x), not just 181. Also, please use “height”, not body height.

Response: Marked with yellow in the “revised manuscript track changes”

2) Line 38, differences (P≤0.05) between exercises, great! Which one is greater?? 

Response: Marked with yellow in the “revised manuscript track changes”

3) Line 41-42, the take-home message is incomplete and unclear. Which exercise is better? What individuals can be beneficial from this?

Response: Line 41-44 is fixed in the “revised manuscript track changes”

Reviewer 2: In the revised Manuscript:

Introduction

General comments:

No hypotheses in the study. 

Response: Line 106-110: Hypotheses are added. 

2) The purpose of the study was not clear. Which 12 different types of exercises should be more specific. 

Response: Fixed aim of the study.

Minor issues:

Line 50-52, many additional….be performed. This is a run-on sentence. 

Response: Line 51-52: Run-on sentence is fixed.

Line 53, prime movers. Since the authors mentioned “much higher”, can the authors report % differences between suspension and standard push-ups in the references cited. 

Response: Line 53: We did not want to give a single percentage value because different studies have different percentages.

Line 60, [5] or (5)? Also, please check the format of this in-text citation. 

Response: Line 60: [5] is corrected.

Line 63-68, the current study did not examine oxygen consumption or RER, this paragraph is not necessary, please delete it. 

Response:Line 63-68: Deleted.

Line 70-71, Differences …. variances. Run-on sentence again, please delete or complete the sentence Response: Line 70-71: Deleted.

Line 82, no significant increase in muscle activation (citations?). 

Response: Line 81: Added citations.

METHODS

General comments:

1) I recommend the authors provide pictures of each exercise, that will help readers understand what exercises performed in the present study.

Response: Added Fig 1.

2) The authors should create a section for MVIC and 1-RM, not included in the “participants” section.

Response: Create new paragraph and deleted from participant section

Minor issues:

Line 112, What did the authors mean recreationally athletes?? Athletes should be professional.

Fixed.

Line 113, height and one decimal.

Response: Line 116 (113 for reviewer) Fixed.

Line 114, how did the authors measure height? Form BIA? 

Response: Line 117-118 (Line 114 for reviewer): Fixed.

Line 120, the authors mentioned the participants perform TRX inverted row and push-up in their training programs. How many years of experience in performing TRX?

Response: Line 120: Fixed.

Line 128-129, MVIC and 1-RM on what exercises?? 

Response: Line 208-242: Fixed (Line 128-129, MVIC and 1-RM on what exercises)

Line 130-131, what exercises did the participants perform in the third visit? 

Response: Line 243-243. Fixed (Line 130-131, what exercises did the participants perform in the third visit?)

Line 147-148, on the belly of what muscles???

Response: Line 153-155: Fixed (belly of what muscles?)

RESULTS

General comments:

1) The authors should create a table for participants’ characteristics with MVIC, 1-RM, and training experience (years).

2) Whenever the authors mentioned differences, the authors should provide if there was a significant difference as well as relationship. Also, the authors should report which one was greater if there is a significant difference.

Minor issues:

Line 234, significant differences, great! Which one was greater?

Line 236, were the differences significant?

Line 238, again, were the differences significant? If yes, which one was greater?

Line 254 and 256, significantly differences, great! Which one was greater?

Line 271, found relationships…? Significant? Positive or negative??

Response: 

1) Created Table 1.

2) Fixed (with yellow marker)

Line 234-236-238-254-271: Fixed (with yellow marker)

DISCUSSION

General comments: The discussion is too long and the authors should create a paragraph to include take-home messages in the last section of the discussion

Response: Fixed. (With yellow marker)

PRACTICAL APPLICATIONS

Line 430-431, what does this sentence mean? “it is possible to exercise with an intensity corresponding to 70% of 1-RM during isotonic suspension exercise.”

What are the authors trying to tell readers??

Line 433-434, I do not understand “further contribute to the future”??

Line 434, should be relative strength “ratio”?

The authors mentioned more unstable does not mean more activation than traditional variations. Why the authors mentioned may achieve more activations when the feet on the device variation?

Response: Fixed (With yellow marker).

---

## [Decision Letter · Decision Letter 1]

3 Nov 2022

PONE-D-22-17523R1

Can Different Variation of Suspension Exercises Provide Adequate Loads and Muscle Activations for Upper Body Training?

PLOS ONE

Dear Dr. Akşit,

Thank you for submitting your manuscript to PLOS ONE. After careful consideration, we have decided that your manuscript does not meet our criteria for publication and must therefore be rejected.

Specifically:

I agree with our reviewers that this is an interesting paper. And I do see there are some improvements in the revised paper. However, as one of the reviewers also mentioned, that the Discussion part is largely repeating the results, and this was still evident in the revised manuscript. 

I am sorry that we cannot be more positive on this occasion, but hope that you appreciate the reasons for this decision.

Kind regards,

Xin Ye, Ph.D.

Academic Editor

PLOS ONE

Reviewers' comments:

Reviewer's Responses to Questions

**Comments to the Author**

1. If the authors have adequately addressed your comments raised in a previous round of review and you feel that this manuscript is now acceptable for publication, you may indicate that here to bypass the “Comments to the Author” section, enter your conflict of interest statement in the “Confidential to Editor” section, and submit your "Accept" recommendation.

Reviewer #1: (No Response)

Reviewer #2: All comments have been addressed

2. Is the manuscript technically sound, and do the data support the conclusions?

Reviewer #1: Partly

Reviewer #2: Yes

3. Has the statistical analysis been performed appropriately and rigorously? 

Reviewer #1: Yes

Reviewer #2: Yes

4. Have the authors made all data underlying the findings in their manuscript fully available?

Reviewer #1: Yes

Reviewer #2: Yes

5. Is the manuscript presented in an intelligible fashion and written in standard English?

Reviewer #1: No

Reviewer #2: Yes

6. Review Comments to the Author

Reviewer #1: I appreciate the hard work in making the corrections in the revised version. However, I consider the authors where not thorough in revising the English grammar. Multiple parts including the revised sessions do not read in standard scientific English.

Reviewer #2: After revision, the article is improved significantly and very clear.

I do not have any further questions.

7. PLOS authors have the option to publish the peer review history of their article (what does this mean?). If published, this will include your full peer review and any attached files.

Reviewer #1: No

Reviewer #2: No

- - - - -

---

## [Author Response · Author response to Decision Letter 1]

22 Feb 2023

Response to Reviewer 1: English grammatical errors and flow have been corrected. by a professional native English assistance. In this process, professional native English assistance was received from “Editage”.

 Rebuttal Letter

Dear Editor,

The events that have occurred from the beginning stages of our study are as follows:

When the initial manuscript was submitted to your journal, it went through the editor's review and then proceeded to the reviewer evaluation process. As a result, two of your reviewers have requested that we make some corrections. In response, the authors have prepared a detailed revision, and all requested changes have been made.

After submitting the revised manuscript, one of the reviewers provided a positive decision, but the other reviewer indicated that the article was insufficient in terms of English language and requested further revisions. The revisions were made and the manuscript was resubmitted to you for a second time. However, in the final stage, while the first reviewer found the manuscript to be satisfactory, the second reviewer stated that the English language revisions were insufficient, and the manuscript was rejected. In response, the authors would like to exercise their right to appeal, as there were no other issues with the manuscript that were communicated to us by your team. 

After our appeal was accepted by your team, we have corrected all English grammatical errors and flow in the manuscript with the help of a professional native English assistance from "Editage". This has addressed all the concerns raised by the reviewers.

We have appealed the decision to reject our manuscript solely based on English language issues, and have resubmitted it to you for further consideration. Considering the positive aspects of our manuscript, we kindly request that you evaluate it again, going through the necessary procedures. We eagerly await your positive response.

---

## [Decision Letter · Decision Letter 2]

15 Apr 2023

PONE-D-22-17523R2Can Different Variations of Suspension Exercises Provide Adequate Loads and Muscle Activations for Upper Body Training?PLOS ONE

Dear Dr. Akşit,

Thank you for submitting your manuscript to PLOS ONE. After careful consideration, we feel that it has merit but does not fully meet PLOS ONE’s publication criteria as it currently stands. Therefore, we invite you to submit a revised version of the manuscript that addresses the points raised during the review process.

The latest revision has been assessed by an independent reviewer and their minor comments are included at the end of this email.  

We look forward to receiving your revised manuscript.

Kind regards,

Xin Ye, Ph.D.

Academic Editor

PLOS ONE

Journal Requirements:

2. You indicated that ethical approval was not necessary for your study. We understand that the framework for ethical oversight requirements for studies of this type may differ depending on the setting and we would appreciate some further clarification regarding your research. Could you please provide further details on why your study is exempt from the need for approval and confirmation from your institutional review board or research ethics committee (e.g., in the form of a letter or email correspondence) that ethics review was not necessary for this study? Please include a copy of the correspondence as an "Other" file. Please note that the current evidence provided as an 'Other' file is not suitable and we require confirmation from your institutional review board or research ethics committee.

Additional Editor Comments (if provided):

Reviewers' comments:

Reviewer's Responses to Questions

**Comments to the Author**

1. If the authors have adequately addressed your comments raised in a previous round of review and you feel that this manuscript is now acceptable for publication, you may indicate that here to bypass the “Comments to the Author” section, enter your conflict of interest statement in the “Confidential to Editor” section, and submit your "Accept" recommendation.

Reviewer #3: All comments have been addressed

2. Is the manuscript technically sound, and do the data support the conclusions?

Reviewer #3: Yes

3. Has the statistical analysis been performed appropriately and rigorously? 

Reviewer #3: Yes

4. Have the authors made all data underlying the findings in their manuscript fully available?

Reviewer #3: Yes

5. Is the manuscript presented in an intelligible fashion and written in standard English?

Reviewer #3: Yes

6. Review Comments to the Author

Reviewer #3: Comments to the Authors

Thanks for your responses to reviewers’ comments. Overall, I think the research question (topic) is clear and the flow of the manuscript is well written. However, several parts of the manuscript may need to be improved, reorganized or reconsidered, as shown below.

Abstract

1. Line 25 and 42: Authors used two different terms ‘traditional’ and ‘conventional’ throughout entire manuscript. Please unify in one word.

2. Line 42-43: It state ‘FD variations of suspension exercises can be more effective than FG variations, …’. I wonder what is more effective in FD variations? Please mention it.

Introduction

3. Page 3, Line 54-61: Authors mentioned about pulley suspension device. Did you use pulley system for this study? If not, is there no big difference with or without the pulley system? I am not sure why this explanation is necessary, if authors have not used pulley.

4. Page 4, Line 73: Which muscle activation was not significantly increased?

5. Page 5, Line 96-99: Are there any reasons for the hypotheses in this study? If so, please add a short description leading to your hypotheses.

Methods

6. Page 5, Line 105: Please provide more information about participant’s sport.

7. Page 5, Line 109-110: It state ‘All participants had a history of resistance exercise for at least 1 year and were right-handed.’ The participants are athletes, so I believe this description doesn't seem appropriate. Please provide other information, such as how many years they have been and what sports they have.

8. Page 6, Line 129: What is SENIAM stands for?

9. Page 6, Line 131: please add ‘and’ between isometric and isotonic.

10. Page 7, Line 142: How to set 60 bpm? Use metronome? If so, please provide information of metronome.

11. Page 7, Line 147: How many participant were excluded?

12. Page 7, Line 149: Please changes ‘exercises’ to ‘suspension exercises’

13. Page 7, Line 156: How do authors know if it's parallel to the ground every time? And when I look at Figure 1, I think the posture is not parallel. In addition, please provide entire body from foot to shoulder for all picture.

14. Page 8, Line 171: For MVIC measurement, did authors use same elbow angle for BP?

15. Page 8, Line 179: What is MIIC? Is it MVIC?

16. Page 8, Line 183: What is TB?

17. Page 9, Line 189: What is TT?

18. Page 9, Line 211: Why the authors use only 3 second of plateau? Not entire 10 seconds?

19. Page 11, Line 238: Please provide detail information about one-way ANOVA. One-way ANOVA for 4 different muscle groups?

Results

20. Page 11, Line 250-253: I believe authors does not need to repeat the same content as the table. Line 250 to 253 is exact same to Table.

Discussion

21. Page 15, Line 319: Please change primary to prime

22. Page 16, Line 327-328: Did the authors measure center of gravity during exercise? If not, please add citation for this sentence.

23. Page 18, Line 378: What does the ‘both exercises’ mean in isometric LBR exercise?

Reference

24. References list are presented in different formats or have missing information (e.g., references # 2, 12, and 15). Please carefully check and revise.

7. PLOS authors have the option to publish the peer review history of their article (what does this mean?). If published, this will include your full peer review and any attached files.

Reviewer #3: No

---

## [Author Response · Author response to Decision Letter 2]

22 May 2023

Editor:

1. Please ensure that you refer to Figure 2 in your text as, if accepted, production will need this reference to link the reader to the figure.

Authors: Fixed this issue.

Editor:

2. We note that you have provided funding information that is currently declared in your Funding Statement. However, funding information should not appear in any areas of your manuscript. We will only publish funding information present in the Funding Statement section of the online submission form. Please remove any funding-related text from the manuscript.

Authors: Deleted funding information from the manuscript.

---

## [Editor Report · Decision Letter 3]

24 Jul 2023

PONE-D-22-17523R3Can Different Variations of Suspension Exercises Provide Adequate Loads and Muscle Activations for Upper Body Training?PLOS ONE

Dear Dr. Akşit,

Thank you for submitting your manuscript to PLOS ONE. After careful consideration, we feel that it has merit but does not fully meet PLOS ONE’s publication criteria as it currently stands. Therefore, we invite you to submit a revised version of the manuscript that addresses the points raised during the review process.

The authors might have forgotten to address Review 3's comments from the previous round, here I am attaching it here:Comments to the Authors

Thanks for your responses to reviewers’ comments. Overall, I think the research question (topic) is clear and the flow of the manuscript is well written. However, several parts of the manuscript may need to be improved, reorganized or reconsidered, as shown below.

Abstract

1. Line 25 and 42: Authors used two different terms ‘traditional’ and ‘conventional’ throughout entire manuscript. Please unify in one word.

2. Line 42-43: It state ‘FD variations of suspension exercises can be more effective than FG variations, …’. I wonder what is more effective in FD variations? Please mention it.

Introduction

3. Page 3, Line 54-61: Authors mentioned about pulley suspension device. Did you use pulley system for this study? If not, is there no big difference with or without the pulley system? I am not sure why this explanation is necessary, if authors have not used pulley.

4. Page 4, Line 73: Which muscle activation was not significantly increased?

5. Page 5, Line 96-99: Are there any reasons for the hypotheses in this study? If so, please add a short description leading to your hypotheses.

Methods

6. Page 5, Line 105: Please provide more information about participant’s sport.

7. Page 5, Line 109-110: It state ‘All participants had a history of resistance exercise for at least 1 year and were right-handed.’ The participants are athletes, so I believe this description doesn't seem appropriate. Please provide other information, such as how many years they have been and what sports they have.

8. Page 6, Line 129: What is SENIAM stands for?

9. Page 6, Line 131: please add ‘and’ between isometric and isotonic.

10. Page 7, Line 142: How to set 60 bpm? Use metronome? If so, please provide information of metronome.

11. Page 7, Line 147: How many participant were excluded?

12. Page 7, Line 149: Please changes ‘exercises’ to ‘suspension exercises’

13. Page 7, Line 156: How do authors know if it's parallel to the ground every time? And when I look at Figure 1, I think the posture is not parallel. In addition, please provide entire body from foot to shoulder for all picture.

14. Page 8, Line 171: For MVIC measurement, did authors use same elbow angle for BP?

15. Page 8, Line 179: What is MIIC? Is it MVIC?

16. Page 8, Line 183: What is TB?

17. Page 9, Line 189: What is TT?

18. Page 9, Line 211: Why the authors use only 3 second of plateau? Not entire 10 seconds?

19. Page 11, Line 238: Please provide detail information about one-way ANOVA. One-way ANOVA for 4 different muscle groups?

Results

20. Page 11, Line 250-253: I believe authors does not need to repeat the same content as the table. Line 250 to 253 is exact same to Table.

Discussion

21. Page 15, Line 319: Please change primary to prime

22. Page 16, Line 327-328: Did the authors measure center of gravity during exercise? If not, please add citation for this sentence.

23. Page 18, Line 378: What does the ‘both exercises’ mean in isometric LBR exercise?

Reference

24. References list are presented in different formats or have missing information (e.g., references # 2, 12, and 15). Please carefully check and revise.==============================

We look forward to receiving your revised manuscript.

Kind regards,

Xin Ye, Ph.D.

Academic Editor

PLOS ONE
---

## [Author Response · Author response to Decision Letter 3]

25 Aug 2023

Dear Reviewer, 

We want to thank for reviewing process of our manuscript and thank you for your precious time.

All reference of lines represented in “Track Changes” manuscript of revision.

ABSTRACT

Reviewer:

1. Line 25 and 42: Authors used two different terms ‘traditional’ and ‘conventional’ throughout entire manuscript. Please unify in one word. 

Authors:

All terms “traditional” change into “conventional” in all manuscript.

Reviewer:

2. Line 42-43: Its state ‘FD variations of suspension exercises can be more effective than FG variations, …’. I wonder what is more effective in FD variations. Please mention it.

Authors:

Line 43: “FD variations of suspension exercises can be more effective in terms of muscle activations than FG variations, and isotonic suspension exercises increase exercise intensity more than isometric suspension exercises” Corrections applied.

INTRODUCTION

Reviewer

Page 3, Line 54-61: Authors mentioned about pulley suspension device. Did you use pulley system for this study? If not, is there no big difference with or without the pulley system? I am not sure why this explanation is necessary if authors have not used pulley.

Authors

Line 57-61: Correction applied. Here, we have only tried to give information about the fact that suspension devices may have other forms and also instability may vary. We did not try to give information about comparing pulley system with our study. If this correction does not fulfil your point of view, surely, we can change it again.

Reviewer

Page 4, Line 73: Which muscle activation was not significantly increased?

Authors

Line 74: In prime mover muscles. Correction applied.

Reviewer

Page 5, Line 96-99: Are there any reasons for the hypotheses in this study? If so, please add a short description leading to your hypotheses.

Authors:

Line 85-86-87-88: These explanations put into manuscript. If this correction does not fulfil your point of view, surely, we can change it again.

METHODS

Reviewer

Page 5, Line 105: Please provide more information about participant’s sport.

Authors

Line 105-106-107. Correction applied.

Reviewer

Page 5, Line 109-110: It state ‘All participants had a history of resistance exercise for at least 1 year and were right-handed.’ The participants are athletes, so I believe this description doesn't seem appropriate. Please provide other information, such as how many years they have been and what sports they have.

Authors

Line 109: We tried to give all the information about the athletes. We have revised the sentence order to meet the meaning.

Reviewer

Page 6, Line 129: What is SENIAM stands for?

Authors

Line 134-135: Correction applied. We tried to give information about SENIAM with long version of word.

Reviewer

Page 6, Line 131: please add ‘and’ between isometric and isotonic.

Authors

Correction was applied.

Reviewer

Page 7, Line 142: How to set 60 bpm? Use metronome? If so, please provide information of metronome.

Authors

Line 150: Correction was added.

Reviewer 

Page 7, Line 147: How many participants were excluded?

Authors

Line 154: No one excluded from the study. Correction was added.

Reviewer

Page 7, Line 149: Please changes ‘exercises’ to ‘suspension exercises’. 

Authors

Line 157: Correction added.

Reviewer

Page 7, Line 156: How do authors know if it's parallel to the ground every time? And when I look at Figure 1, I think the posture is not parallel. In addition, please provide entire body from foot to shoulder for all pictures.

Authors

Line 163. Correction applied. Unfortunately, these were the best photos we could take with the means at our disposal.

Reviewer

Page 8, Line 171: For MVIC measurement, did authors use same elbow angle for BP?

Author

Line 181-182: Corrections was added.

Reviewer

Page 8, Line 179: What is MIIC? Is it MVIC?

Author

Line 189: Fixed.

Reviewer

Page 8, Line 183: What is TB?

Page 9, Line 189: What is TT?

Authors

Line 193 for “TB” and Line 199 for “TT”: Correction applied.

Reviewer

Page 9, Line 211: Why the authors use only 3 second of plateau? Not entire 10 seconds? 

Authors

Line 222-223: “The first and the last 3 seconds are excluded from the data to prevent the data from being distorted by the small oscillations that may occur in the body” was added.

Reviewer

Page 11, Line 238: Please provide detail information about one-way ANOVA. One-way ANOVA for 4 different muscle groups?

Authors

Line 248-249: Correction added.

RESULTS

Reviewer

Page 11, Line 250-253: I believe authors does not need to repeat the same content as the table. Line 250 to 253 is exact same to Table. 

Authors

All lines were deleted.

DISCUSSION

Reviewer

Page 15, Line 319: Please change primary to prime.

Authors

Checked.

Reviewer

Page 16, Line 327-328: Did the authors measure center of gravity during exercise? If not, please add citation for this sentence.

Authors

Lines 347-349: No, we did not measure. Sentence was fixed. “In the FD variation of isotonic push-ups, the body’s center of gravity moved forward toward the straps and the stabilization decreased further, and PM activation was observed to be higher than that of the BP, which is a more stable exercise.”

Reviewer

Page 18, Line 378: What does the ‘both exercises’ mean in isometric LBR exercise?

Authors

Line 399: Correction was added.

REFERENCES

Reviewer

References lists are presented in different formats or have missing information (e.g., references # 2, 12, and 15). Please carefully check and revise.

Authors

All references checked and corrected. Thank you for your attention.

---

## [Editor Report · Decision Letter 4]

4 Sep 2023

Can Different Variations of Suspension Exercises Provide Adequate Loads and Muscle Activations for Upper Body Training?

PONE-D-22-17523R4

Dear Dr. Akşit,

We’re pleased to inform you that your manuscript has been judged scientifically suitable for publication and will be formally accepted for publication once it meets all outstanding technical requirements.

Kind regards,

Xin Ye, Ph.D.

Academic Editor

PLOS ONE
---

## [Editor Report · Acceptance letter]

13 Sep 2023

PONE-D-22-17523R4 

Can Different Variations of Suspension Exercises Provide Adequate Loads and Muscle Activations for Upper Body Training? 

Dear Dr. Akşit:

I'm pleased to inform you that your manuscript has been deemed suitable for publication in PLOS ONE. Congratulations! Your manuscript is now with our production department. 

Kind regards, 

on behalf of

Dr. Xin Ye 

Academic Editor

PLOS ONE